# Gene Expression Profile in the Sandhoff Mouse Brain with Progression of Age

**DOI:** 10.3390/genes13112020

**Published:** 2022-11-03

**Authors:** Kshitiz Singh, Brianna M. Quinville, Melissa Mitchell, Zhilin Chen, Jagdeep S. Walia

**Affiliations:** 1Department of Pediatrics, Queen’s University, Kingston, ON K7L 3N6, Canada; 2Centre for Neuroscience Studies, Queen’s University, Kingston, ON K7L 3N6, Canada

**Keywords:** Sandhoff disease, qPCR, gene expression, *Hexb*, *Wfdc17*, *Ccl3*, *Lyz2*, *Fa2h*, *Mog*, *Ugt8a*

## Abstract

Sandhoff disease (SD) is a fatal neurodegenerative disorder belonging to the family of diseases called GM2 Gangliosidosis. There is no curative treatment of SD. The molecular pathogenesis of SD is still unclear though it is clear that the pathology initiates with the build-up of ganglioside followed by microglial activation, inflammation, demyelination and apoptosis, leading to massive neuronal loss. In this article, we explored the expression profile of selected immune and myelination associated transcripts (*Wfdc17*, *Ccl3*, *Lyz2*, *Fa2h*, *Mog* and *Ugt8a*) at 5-, 10- and 16-weeks, representing young, pre-symptomatic and late stages of the SD mice. We found that immune system related genes (*Wfdc17*, *Ccl3*, *Lyz2*) are significantly upregulated by several fold at all ages in *Hexb*-KO mice relative to *Hexb*-het mice, while the difference in the expression levels of myelination related genes is not statistically significant. There is an age-dependent significant increase in expression of microglial/pro-inflammatory genes, from 5-weeks to the near humane end-point, i.e., 16-week time point; while the expression of those genes involved in myelination decreases slightly or remains unchanged. Future studies warrant use of new high-throughput gene expression modalities (such as 10X genomics) to delineate the underlying pathogenesis in SD by detecting gene expression changes in specific neuronal cell types and thus, paving the way for rational and precise therapeutic modalities.

## 1. Introduction

Sandhoff Disease (SD), Tay Sachs (TSD) and AB variant are lysosomal storage disorders that belong to a family of diseases called GM2 Gangliosidosis. They are fatal neurodegenerative disorders that are inherited in an autosomal recessive manner, and are caused by a genetic mutation that results in an accumulation of the incompletely metabolized GM2 mono-sialoganglioside in the lysosome of cells, especially in the central nervous system (CNS) [1,2]. The occurrence of SD is approximately 1 in 300,000 in the general population, but is increased in certain populations [3]. 

GM2 Gangliosidosis occurs due to a deficiency of the lysosomal enzyme β-Hexosaminidase-A (Hex A) or the GM2A Activator Protein. The X-ray crystallographic structure of Hex A has revealed that it is an αβ heterodimer [4]. The α and β subunits are encoded by *HEXA* and *HEXB* genes, respectively, [5]. Deficiency of the *HEXB* gene results in SD, whilst deficiency of the *HEXA* and GM2A Activator (GM2A) results in TSD and AB variant respectively. SD occurs in 3 forms: infantile, juvenile and adult form; with the infantile form being the most common and the most severe due to the least amount of enzyme activity [6,7]. 

There is no curative treatment for GM2 gangliosidosis and in several animal models, treatment modalities like enzyme replacement, substrate restriction, glucose analogs and anti-inflammatory agents have shown variable benefits [8,9]. Gene therapy and small molecule approaches are also being explored to alleviate the symptoms and lifespan of these diseases [10,11,12]. Further, to guide the development of rational and precise therapeutic approaches, specific delineation of molecular pathology in these diseases is vital.

A number of research groups have explored into the pathogenesis of GM2 Gangliosidosis, as well as its commonality among other lysosomal storage disorders. Some of the main features of the pathology include the build-up of ganglioside, microglial activation, inflammation, demyelination and apoptosis, leading to massive neuronal loss [13,14,15]. A recent microarray study on brain tissues by Ogawa and colleagues (2018) identified a number of differentially expressed genes related to inflammation and demyelination in 4-week old SD mice [16]. Here, we attempted validating a subset (Table 1) of these putative differentially regulated genes in a SD mouse model. We explored the expression profile of selected immune and myelination associated transcripts at 5-, 10- and 16-weeks, representing young, pre-symptomatic and late stages of the SD mice, to delineate the progression of the changes with age and localize their expression pattern in cerebral cortex and cerebellum, the two most affected parts of the brain. 

## 2. Materials and Methods

### 2.1. Animals

All experiments were performed in accordance with Queen’s University Animal Care Committee (2017-1707). The animals used in this study were *Hexb*−/− (*Hexb*-KO) Sandhoff disease mouse model (B6;129S4-*Hexb^tm1Rlp^*, Jackson Laboratory, Bar Harbor, ME, USA). This colony is on a 12 h day/night cycle (7 to 7), in Tecniplast Green Line IVC Sealsafe cages, receiving a Laboratory Rodent Diet with water ad libitum. This mouse model abnormally accumulates GM2 and GA2 ganglioside, develop motor defects beginning at about 3 months of age. Without treatment, the defects progressively worsen and *Hexb*-KO mice die by 16 to 18 weeks of age. 

### 2.2. Brain Sample Collection and RNA Extraction

Midsections of the brain (representative cerebral cortex) and cerebellum were stored in RNALater (Invitrogen, Waltham, MA, USA) at −80 °C until all samples were ready for extraction in batch. Extractions were made using GeneJet RNA Purification Kit (Thermo Fisher, Mississauga, ON, Canada) according to manufacturer’s recommendations. Genomic DNA was removed by DNase treatment. RNA was eluted with 50 μL of nuclease-free water. RNA Quality and concentration was measured using a Nanodrop 1000 spectrophotometer (Thermo Fisher Scientific, Waltham, MA, USA). RNA samples with A260/280 between 1.8–2.1 were used for cDNA synthesis.

### 2.3. qPCR

cDNA was synthesized using the Qiagen QuantiTect Reverse Transcription Kit (Qiagen, Québec City, QC, Canada) according to manufacturers protocols. The concentration of the cDNA was measured using Nanodrop 1000 spectrophotometer and was normalized to 100 ng/µL. 200 nanogram of cDNA per well was used for qPCR. was TaqMan probes (Appendix A, Thermo Fisher, Mississauga, ON, Canada) and Thermo Fisher’s TaqMan Fast Advanced Master Mix (#4444557, Thermo Fisher Scientific) were used for RT-PCR to determine gene expression using Quant Studio 3 RT-PCR instrument (Thermo Fisher, Mississauga, ON, Canada). Plate set up incorporated a randomized design, in order to reduce plate bias. The samples were therefore distributed across 2, 96-well plates, for each brain section, with 3 time points per plate. PCR conditions were followed according to manufacturer’s recommendations (TaqMan Fast Advanced Master Mix user guide Publication Number MAN0025706, Revision A.0) The 10 Heterozygote (het; *Hexb*−/+) and 10 knockout (KO) samples were randomized across the plates; therefore, there were 5 het and 5 KO samples per plate. The samples were pipetted using an electronic multichannel pipette in triplicate. 

### 2.4. Protein Functional Association Network Analysis

STRING database (Creative Commons CC BY 4.0 license) was used for predicting functional associations between proteins [17]. Briefly, this database integrates all known and predicted associations between proteins. Both physical and functional associations are considered in the networks. Single protein name of one of the tested genes (*Hexb*, *Wfdc17*, *Ccl3*, *Lyz2*, *Fa2h*, *Mog or Ugt8a*) was searched in STRING database. Organism ‘Mus musculus’ was chosen from the list. Then, the database summarized the network of predicted associations for the searched protein. Functional protein annotations from STRING database are assembled in the tables.

### 2.5. Data Analysis and Statistics

Quantitative PCR (qPCR) data analysis was done based on the Taylor et al. [18] and MIQE (Minimum Information for Publication of Quantitative Real-Time PCR Experiments) guidelines. Briefly, Mean Ct (threshold cycle) of three technical replicates for each target was calculated. *B-Actin* was used as the reference gene. Brain (cerebral cortex or cerebellum) samples from 5-weeks old *Hexb*−/+ (*Hexb*-het) mice were considered control group. Average Ct for each target in the control group was calculated. The relative difference (deltaCt) between the average Ct for the control group and the mean Ct per individual sample within each target was determined. Relative quantity (fold change) was calculated using deltaCt i.e 2^deltaCt^. Then, normalized expression was determined by dividing relative quantity of target gene by the relative quantity of the reference gene. Statistical data analysis and graph plots were made using GraphPad Prism version 9.3.1 for Windows, GraphPad Software, San Diego, CA, USA. 

## 3. Results

### 3.1. Wfdc17

In *Hexb*-KO cerebellum and cerebral cortex, the expression of *Wfdc17* was significantly higher (*p* value < 0.05) than in *Hexb*-het mice. The pattern of expression differed between the cerebellum and cerebral cortex. In the cerebellum, *Wfdc17* expression increased with age. However, in the *Hexb*-KO cerebral cortex the expression remained relatively unchanged with age, albeit levels were higher in *Hexb*-KO as compared to *Hexb*-het cohort (Figure 1).

### 3.2. Ccl3

*Ccl3* expression in *Hexb*-KO cerebellum and cerebral cortex remained higher than in *Hexb*-het at 5, 10 and 16 weeks of age (*p* value < 0.05). Moreover, there was a tendency of increase in *Ccl3* expression in both cerebellum and cerebral cortex of *Hexb*-KO mice with age. The slope of rise in *Ccl3* expression between 5 to 16 weeks was higher in cerebellum, as compared to the cerebral cortex, as shown in Figure 2.

### 3.3. Lyz2

Expression of *Lyz2* in the cerebellum and cerebral cortex showed a similar pattern as of *Wfdc17* in *Hexb*-KO and *Hexb*-het mice. In *Hexb*-KO cerebellum and cerebral cortex, the expression of *Lyz2* was significantly higher (*p* value < 0.05) than in *Hexb*-het mice. In the cerebral cortex, there was not a drastic change in the expression of *Lyz2* with age; however, in the cerebellum, *Lyz2* expression showed a rising trend with age. At 16 weeks of age, in *Hexb*-KO cerebellum, the *Lyz2* expression was 3.2 times relative to 5-weeks expression. In the cerebellum of *Hexb*-het mice, there was a slight rising trend of *Lyz2* expression between 5 to 16 weeks, while in the cerebellum of *Hexb*-KO mice there was a steeply rising trend of Lyz2 gene expression (Figure 3).

### 3.4. Fa2h

In the cerebellum and cerebral cortex, *Fa2h* expression levels in *Hexb*-KO were slightly lower than in *Hexb*-het for all ages. However, these levels were significant only at 10 weeks in the cerebellum and 10 weeks and 16 weeks in the cerebral cortex. 

### 3.5. Mog

In the cerebellum, there was a downward trend in expression in *Hexb*-het and *Hexb*-KO mice. The analogous tendency of *Mog* expression is detectable in the cerebral cortex as well (Figure 4).

### 3.6. Ugt8a

In *Hexb*-KO cerebellum and cerebral cortex, *Ugt8a* gene showed a similar level of expression as in *Hexb*-het and the expression remained relatively unchanged between 5 to 16 weeks age for this gene, as shown in Figure 5.

### 3.7. Protein Association Network Analysis

Further, we performed protein association network analysis using a string database for *Hexb*, *Wfdc17*, *Ccl3*, *Lyz2*, *Fa2h*, *Mog* and *Ugt8a (*Figure 6). In Figure 6a, *Hexb* is predicted to have a functional association with several genes, namely, *Hexa*, *Gns*, *Glb1*, *Naga*, *Gla*, *Renbp*, *Chit1*, *Chil6*, *Nagk and Chia1.* Most genes of the *Hexb* network are involved in the degradation of GM2 gangliosides and chitin (Table 2).

For Wfdc17 functional association network (Figure 6b), most genes are associated with the immune system related pathways, with functions related to antigen uptake and cell–cell interaction, in macrophages, microglia and neutrophils (Figure 6b, Table 3).

Ccl3 is a chemokine and has inflammatory and chemokinetic characteristics. Its protein association network (Figure 6c, Table 4), primarily consists of other chemokine receptors, interleukins, GM-CSF (granulocyte-monocyte colony-stimulating factor) and tumor necrosis factor. This immune system activation was likely the result of gangliosides accumulation.

Lyz2 was found to be associated with Lyz1. Their activity is primarily associated with monocyte-macrophage system. Fa2h protein is primarily responsible for fatty acid hydroxylation and generating hydroxylated sphingolipids (Figure 6d, Table 5). Fa2h protein association network (Figure 6e, Table 6) consists of unsaturated fatty acids generation, lipid synthesis, glucosylceramide catalysis, intracellular vesicle mobility (Dync1li2) and apoptosis (Pla2g6). 

The protein association network of Mog (Figure 6f, Table 7) consists of proteins involved in maternal and paternal gene imprinting (by DNA methylation and other epigenetic modifications; Zfp57), CNS osmoregulation (Aquaporin-4), interleukin, immune system associated receptors and co-receptors (Cd209d, Cd4) and other myelin-associated proteins.

The protein association network of Ugt8a consists of Cers3, Smpd4, Sgpp2, Ugcg, Cerk, Sgms1, Cers2 and Degs2 (Figure 6g, Table 8). Most of these genes are associated with cerebrosides and sphingosines metabolism.

## 4. Discussion

SD and TSD are fatal neurodegenerative diseases that result from the build-up of GM2 Gangliosides in the lysosomes of neurons. The molecular changes occurring in GM2 Gangliosidosis are still not well understood [13,14]. In this study, we studied the expression of selected genes responsible for inflammation and myelination (*Wfdc17*, *Ccl3*, *Lyz2*, *Fa2h*, *Mog* and *Ugt8a)* for an improved understanding of the disease process. *Wfdc17*, *Ccl3* and *Lyz2* are primarily associated with immune system-related pathways, while *Fa2h*, *Mog* and *Ugt8a* are associated with myelination. Microglial activation, inflammation and associated demyelination are known causative factors for this disease and the genes for inflammation and demyelination have been found to be differentially regulated in the asymptomatic phase of the SD mouse model [16,19]. However, the temporal relationship of molecular mechanisms which lead to the development of symptoms is unclear. Therefore, in this study, we measured the expression of genes at early (5 weeks), intermediate (10 weeks) and late phase (16 weeks) of the disease in an attempt to establish temporal changes in expression of certain genes.

We found that immune system related genes (*Wfdc17*, *Ccl3*, *Lyz2)* are significantly upregulated, by several fold, at all ages in *Hexb*-KO mice relative to *Hexb*-het mice, while the difference in the expression levels of myelination related genes are not statistically significant. This suggests that the primary cause of pathology lies within immune system related genes. Moreover, the immune related gene expression changes are more pronounced in cerebellum as compared to the cerebral cortex. The levels of myelination-associated genes appear to have a lower expression trend in *Hexb*-KO mice, as compared to *Hexb*-het mice, however, this difference is not statistically significant in our study. 

Microglia are the resident macrophages of the brain [20]. In the disease state, microglia respond to infections, disease, and injury in their primary role of phagocytosis and cytokine secretion. However, over time, rather than mitigating neuronal damage, they precipitate an inflammatory response. Pro-inflammatory cytokines released by microglia lead to further microglial activation, creating a positive feedback loop. This loss of “checkpoints” in neuroinflammatory pathways leads to a sustained inflammatory response and eventually neurodegeneration [21,22]. Microglial activation has been shown to precede neuronal death in SD mice, driven by lipid storage-induced microglia activation [13]. It has also been shown that lipid storage seen in GM2 Gangliosidosis may impair phagocytosis, leading to the extended release of toxins by microglia; whilst the phagocytosis of apoptotic neurons may actually be a trigger for pathogenesis [23]. The GM2 and other gangliosides accumulation triggers autoimmune responses, likely, through the interaction with Fcr-γ [16].

Ccl3, also termed macrophage inflammatory protein 1 α (MIP-1-α) is a chemokine that is produced endogenously by microglia, mediating an inflammatory response; a key feature of GM2 Ganglioside pathogenesis [24,25,26]. Davetelis and colleagues 2004, showed that when this gene is knocked out in SD mice, the disease is ameliorated [21]. Chemokines are thought to activate the NF-kappa B transcription factor, which itself regulates chemokines, creating a positive feedback loop, seen in autoimmune inflammation [27,28]. *Ccl3* expression increases with the severity of the disease in *Hexb*-KO mouse models. 

*Lyz2* (also known as *Lyz M*) expression has also increased in the *Hexb*-KO cohort compared to the *Hexb*-het cohort at each time point, within a midsection of the brain as well as in the cerebellum. An early study into Lyz2, by Cross and colleagues (1998), characterized the *Lyz2* gene and showed that Lyz2 is expressed in a tissue-dependent manner and has different expression profiles according to cell type, suggesting that Lyz2 is a potential marker of macrophages [29,30]. Lyz2 murine models have also been used to study the different populations of innate immune cells, in order to help understand disease progression, such as in multiple sclerosis, and the effect of immune cell response in its absence, highlighting their significance as a marker of microglial activation [31,32]. Expression of *Lyz2* shows a slight downward trend in the cerebral cortex for *Hexb-hets*, while in the cerebellum, the expression has a slight upward slope (Figure 3). Microglia, in their homeostatic role, are thought to support myelinogenesis in the developmental brain (pre- and postnatally) [33,34]; they are also thought to continue to support the oligodendrocyte progenitor cells (OPCs) in the adult brain [35]. This increase, therefore, may be considered to reflect patterns of myelination. 

The *Wfdc17* gene, a WAP domain protein, also known as activated microglia/macrophage WAP domain protein (AMWAP), has been shown to be active at the earliest stages of microglial activation, reducing transcripts of pro-inflammatory markers, whilst resulting in upregulation of other microglial markers. It is thought to not only counteract the inflammatory response, but also promote a homeostatic microglial response, as well as reducing neurotoxicity [36,37]. The relative gene expression levels for *Wfdc17* in the cerebellum increase with age in *Hexb*-KO, as compared to *Hexb*-het mice, while in the cerebral cortex the expression levels remain largely unchanged. *Hexb*-KO cerebellum and cerebral cortex show higher levels than in *Hexb*-het, but relative levels are lower in the cerebral cortex. A similar expression pattern is visible with *Lyz2* as well. *Lyz2* and *Wfdc17* are both thought to be regulated by transcription factor NF-kappa B [28]. NF-kappa B is induced by Ccl3, suggesting that *Wdfc17* and *Lyz2* increase following microglial activation after *Ccl3* release. The increased expression of Ccl3 as a marker of microglial activation, Lyz2 as an inflammatory marker, and Wdfc17 as a marker of anti-inflammatory response suggests that from 5-weeks, there is microglial activation and an inflammatory response occurring in both the cerebral cortex and the cerebellum. These genes are all thought to be mediated by the NF-kappa B transcription factor, which itself is induced by Ccl3, regulating chemokines in a positive feedback loop. 

Among the genes related to myelination—*Ugt8A*, *Fa2h* and *Mog*, our data show that in both *Hexb*-KO and *Hexb*-het cerebral cortex and cerebellum, *Fa2h* and *Mog* have a downward trend of expression with age, while *Ugt8a* does not change significantly with age. *Mog* and *Fa2h* expression are consistently lower in *Hexb*-KOs, compared to *Hexb*-hets, the difference is significant at 10-weeks or 12-weeks of age in the cerebral cortex and cerebellum (Figure 4 and Figure 7). For *Mog* and *Fa2h*, it appears that the lack of statistical significance is due to the limited number of mice used in this study.

There is not a significant change between the levels of Ugt8a in the cerebral cortex and cerebellum at any age (Figure 5). The *Ugt8a* gene (UDP Galactosyltransferase 8), also known as UDP-galactose: ceramide galactosyltransferase, and commonly referred to as the CGT gene [38]. However, a murine model deficient in *Ugt8a*, showed impaired axon insulation resulting in body tremors and loss of motor activity. This phenotype is similar to that seen in SD mice [39].

The *Fa2h* gene codes for fatty acid 2-hydroxylase (also known as fatty acid-α-hydroxylase). The expression of *Fa2h* in *Hexb*-KOs remains lower than *Hexb*-hets, however, the difference is not statistically significant. There is also a downward trend of *Fa2h* expression with age. Fatty acid 2-hydroxylase catalyzes fatty acid 2-hydroxylation, synthesizing 2-hydroxy-fatty acids. These fatty acids are reported to be an important component of viable myelin, the absence of which can result in demyelination [40]. Expression of *Fa2h* shows a downward trend in the cerebral cortex and the cerebellum (Figure 7).

*Mog*, or Myelin-Oligodendrocyte Glycoprotein is a minor glycoprotein found on the external lamellae of myelin sheaths and on the surface of myelinating oligodendrocytes. Due to the “external location” of the *Mog* protein in the myelin sheath and the fact that it occurs in the later stages of myelination, it is thought to be important for myelination completion and maintenance [41]. Its location on the surface of myelin sheath makes it a target for antibodies in autoimmune diseases, such as multiple sclerosis [42,43]. The expression of the *Mog* gene in the cerebral cortex and cerebellum of *Hexb*-hets remains higher than in *Hexb*-KOs, however, the difference is not statistically significant (Figure 4), similar to the pattern of the *Ugt8a* gene and the *Fa2h* genes. This aligns with the previous studies of *Mog* expression which suggest that following higher expression a few days after birth, the *Mog* expression is maintained at a lower level for myelin maintenance [41,43]. 

Gene expression related to demyelination, including genes *Ugt8a*, *Fa2h and Mog*, show a general lower level in the *Hexb*-KOs, compared to the *Hexb*-hets. Though, the trend does not appear to be statistically significant, probably due to the lower number of mice used in this study. However, since the levels are lower in KO as compared to the heterozygotes at each time point analyzed, it indicates that other mechanisms associated with inflammatory pathways might also be playing a role. The *Ugt8a* and *Fa2h* have been shown to be vital genes in the formation of galactosylceramides, an important component of myelin [40]. Without them, viable myelin is not synthesized. Mog is also a recognized marker of myelination [44].

Further examination of functional protein networks (Figure 6) has revealed the involvement of other downstream or upstream genes and has shed some light regarding the pathway which might get affected by these interactions. Future experiments will be needed to outline the specific pathways that possibly affect myelination. Current evidence from this study indicates an overwhelming activation of inflammatory pathways; however, the extent of demyelination in the disease process remains unclear.

This study has the limitation of analyzing the restricted number of genes by qPCR. However, we demonstrate that there is an age-dependent significant increase in expression of microglial/inflammatory genes, from 5-weeks to the near humane end-point, i.e., 16-week time point. While the expression of those genes involved in myelination decreases slightly or remains unchanged. In conformance to the previous knowledge of cerebellum pathology, the inflammatory gene expression was found more pronounced in the cerebellum, as compared to the cerebral cortex. This information lays the foundation for future studies which shall target exhaustive, high-throughput gene expression modalities (such as 10X genomics). This will delineate the gene expression changes at different ages to determine expression of genes in specific neuronal cell types and thus further extensively investigate the mechanisms and effects of these inflammatory reactions within the SD brain. Such knowledge will pave the way for better understanding of molecular pathophysiology and may guide rational and precise therapeutic modalities.

## Figures and Tables

**Figure 1 genes-13-02020-f001:**
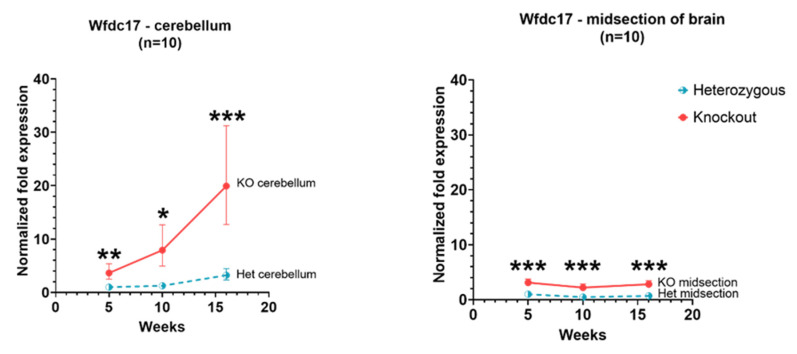
Relative *Wfdc17* expression in the cerebellum and cerebral cortex of *Hexb*-KO Sandhoff disease mouse model (*—*p* value < 0.05; **—*p* value < 0.01; ***—*p* value < 0.001; midsection of brain represents cerebral cortex).

**Figure 2 genes-13-02020-f002:**
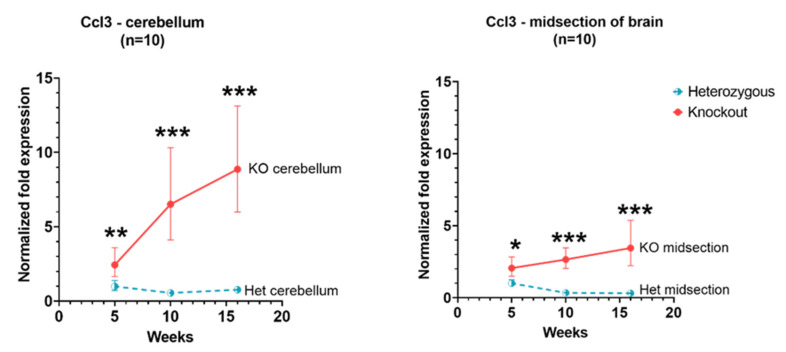
Relative *Ccl3* expression in the cerebellum and cerebral cortex of *Hexb*-KO Sandhoff disease mouse model (*—*p* value < 0.05; **—*p* value < 0.01; ***—*p* value < 0.001; midsection of brain represents cerebral cortex).

**Figure 3 genes-13-02020-f003:**
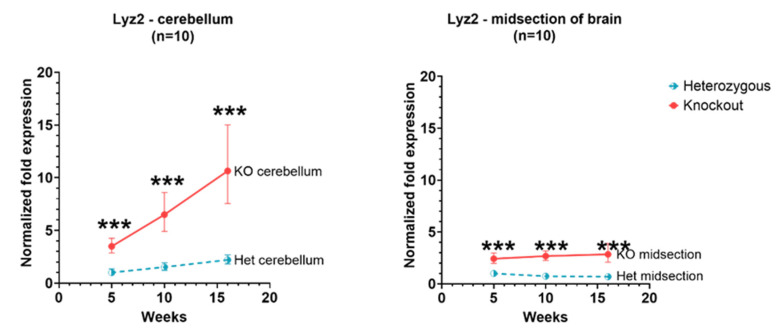
Relative Lyz2 expression in the cerebellum and cerebral cortex of Hexb-KO Sandhoff disease mouse model (***—*p* value < 0.001; midsection of brain represents cerebral cortex).

**Figure 4 genes-13-02020-f004:**
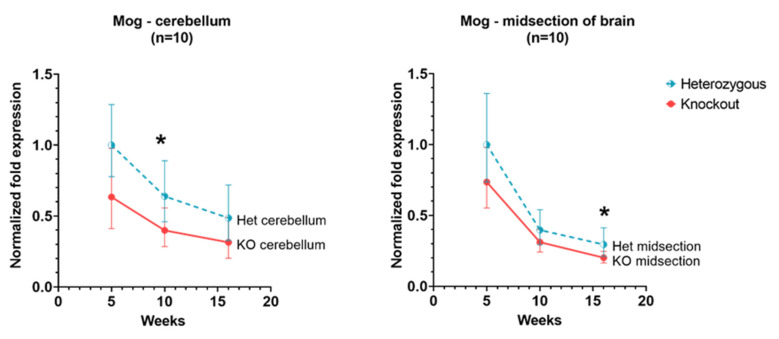
Relative Mog expression in the cerebellum and cerebral cortex of *Hexb*-KO Sandhoff disease mouse model (*—*p* value < 0.05; midsection of brain represents cerebral cortex).

**Figure 5 genes-13-02020-f005:**
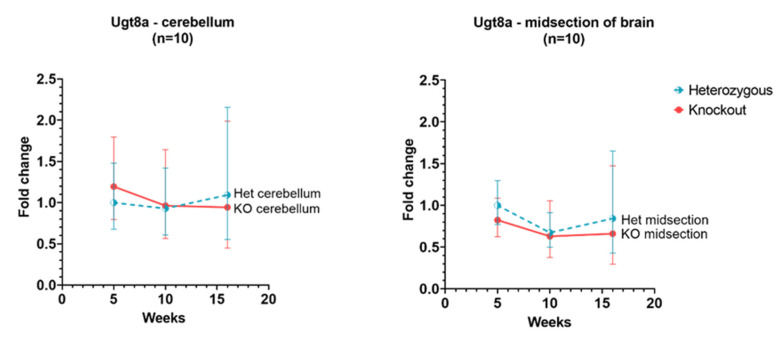
Relative Ugt8a expression in the cerebellum and cerebral cortex of Hexb-KO Sandhoff disease mouse model (midsection of brain represents cerebral cortex).

**Figure 6 genes-13-02020-f006:**
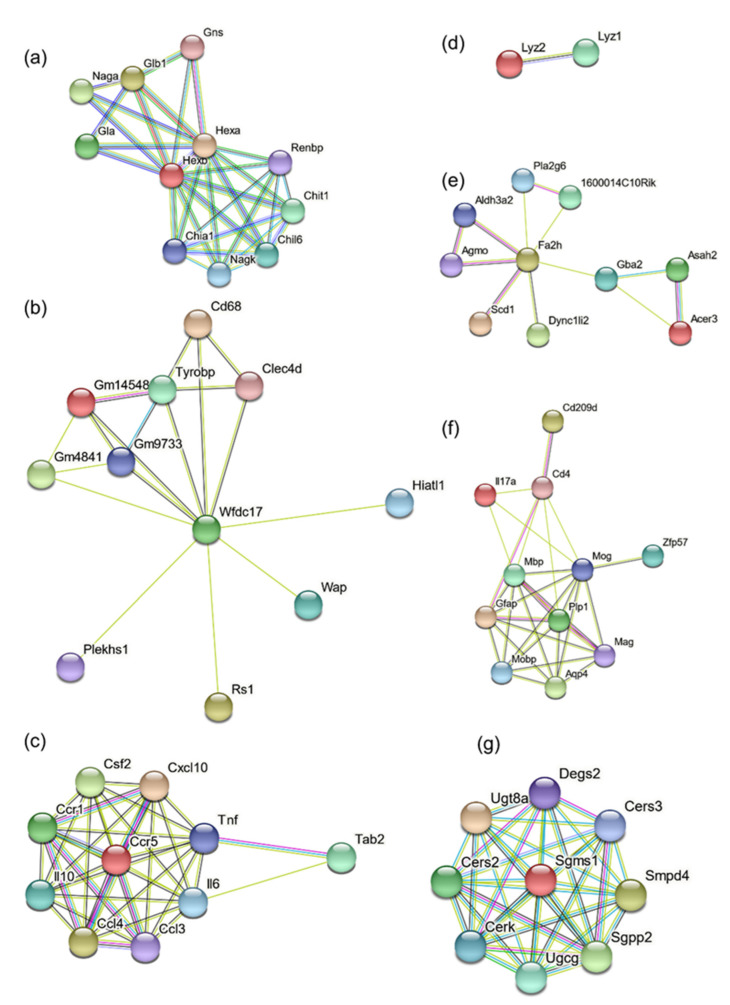
Functional protein association network clusters for (**a**) *Hexb*, (**b**) *Wfdc17*, (**c**) *CCl3*, (**d**) *Lyz2*, (**e**) *Fa2h*, (**f**) *Mog*, and (**g**) *Ugt8a*.

**Figure 7 genes-13-02020-f007:**
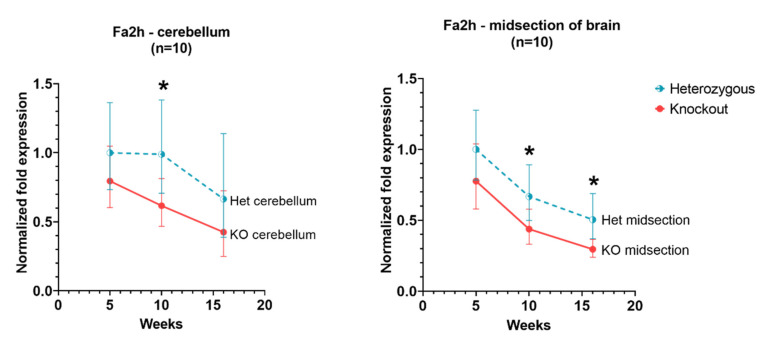
Relative *Fa2h* expression in the cerebellum and cerebral cortex of *Hexb*-KO Sandhoff disease mouse model (*—*p* value < 0.05; midsection of brain represents cerebral cortex).

**Table 1 genes-13-02020-t001:** NCBI IDs and gene function annotation using String database.

NCBI ID	Gene	Annotation
100034251	*Wfdc17*	Activated macrophage/microglia WAP domain protein; WAP four-disulfide core domain 17
20302	*Ccl3*	C-C motif chemokine 3; Monokine with inflammatory, pyrogenic and chemokinetic properties. Has a potent chemotactic activity for eosinophils. Binding to a high-affinity receptor activates calcium release in neutrophils; Belongs to the intercrine β (chemokine CC) family
17105	*Lyz2*	Lysozyme C-2; Lysozymes have primarily a bacteriolytic function; those in tissues and body fluids are associated with the monocyte- macrophage system and enhance the activity of immunoagents. Lyz2 is active against a range of Gram-positive and Gram-negative bacteria. More effective than Lyz1 in killing Gram-negative bacteria. Lyz1 and Lyz2 are equally effective in killing Gram- positive bacteria; Belongs to the glycosyl hydrolase 22 family
338521	*Fa2h*	4-hydroxysphinganine ceramide fatty acyl 2-hydroxylase; Fatty acid 2-hydroxylase; Required for α-hydroxylation of free fatty acids and the formation of α-hydroxylated sphingolipids
17441	*Mog*	Myelin oligodendrocyte glycoprotein; Minor component of the myelin sheath. May be involved in completion and/or maintenance of the myelin sheath and in cell–cell communication. Mediates homophilic cell–cell adhesion
22239	*Ugt8a*	2-hydroxyacylsphingosine 1-β-galactosyltransferase; Catalyzes the transfer of galactose to ceramide, a key enzymatic step in the biosynthesis of galactocerebrosides, which are abundant sphingolipids of the myelin membrane of the central nervous system and peripheral nervous system; Belongs to the UDP-glycosyltransferase family

**Table 2 genes-13-02020-t002:** Functional annotation of HexB protein association network.

Gene	Annotation
*Hexb*	β-hexosaminidase subunit β; Responsible for the degradation of GM2 gangliosides, and a variety of other molecules containing terminal N-acetyl hexosamines, in the brain and other tissues
*Hexa*	β-hexosaminidase subunit α; Responsible for the degradation of GM2 gangliosides, and a variety of other molecules containing terminal N-acetyl hexosamines, in the brain and other tissues
*Gla*	α-galactosidase a; Galactosidase, α
*Naga*	α-N-acetylgalactosaminidase; Removes terminal α-N-acetylgalactosamine residues from glycolipids and glycopeptides. Required for the breakdown of glycolipids (By similarity); Belongs to the glycosyl hydrolase 27 family
*Glb1*	Galactosidase, β 1; β-galactosidase; Cleaves β-linked terminal galactosyl residues from gangliosides, glycoproteins, and glycosaminoglycans
*Gns*	N-acetylglucosamine-6-sulfatase; Glucosamine (N-acetyl)-6-sulfatase; Belongs to the sulfatase family
*Chia1*	Acidic mammalian chitinase; Degrades chitin and chitotriose. May participate in the defense against nematodes, fungi and other pathogens. Plays a role in T-helper cell type 2 (Th2) immune response. Contributes to the response to IL-13 and inflammation in response to IL-13. Stimulates chemokine production by pulmonary epithelial cells. Protects lung epithelial cells against apoptosis and promotes phosphorylation of AKT1.
*Nagk*	N-acetyl-D-glucosamine kinase; Converts endogenous N-acetylglucosamine (GlcNAc), a major component of complex carbohydrates, from lysosomal degradation or nutritional sources into GlcNAc 6-phosphate. Involved in the N-glycolylneuraminic acid (Neu5Gc) degradation pathway. Furthermore, has ManNAc kinase activity
*Chil5*	Chitinase 3-like 3/4; Belongs to the glycosyl hydrolase 18 family
*Chit1*	Chitotriosidase-1; Degrades chitin, chitotriose and chitobiose. May participate in the defense against nematodes and other pathogens (By similarity); Belongs to the glycosyl hydrolase 18 family. Chitinase class II subfamily

**Table 3 genes-13-02020-t003:** Functional annotation of Wfdc17 protein association network.

Gene	annotation
*Wfdc17*	Activated macrophage/microglia WAP domain protein; WAP four-disulfide core domain 17
*Gm9733*	Signal-regulatory protein α/beta1/γ; SIRP β 1 like 2 protein; Predicted gene 9733
*Gm4841*	Interferon-γ-inducible GTPase Ifgga3 protein; Predicted gene 4841
*Gm14548*	Leukocyte immunoglobulin-like receptor; Predicted gene 14548
*Tyrobp*	TYRO protein tyrosine kinase-binding protein; Non-covalently associates with activating receptors of the CD300 family. Cross-linking of CD300-TYROBP complexes results in cellular activation. Involved for instance in neutrophil activation mediated by integrin
*Clec4d*	C-type lectin domain family 4 member D; Functions as an endocytic receptor. May be involved in antigen uptake at the site of infection, either for clearance of the antigen, or for processing and further presentation to T-cells (By similarity)
*Cd68*	Macrosialin; Could play a role in phagocytic activities of tissue macrophages, both in intracellular lysosomal metabolism and extracellular cell–cell and cell-pathogen interactions. Binds to tissue- and organ-specific lectins or selectins, allowing homing of macrophage subsets to particular sites. Rapid recirculation of CD68 from endosomes and lysosomes to the plasma membrane may allow macrophages to crawl over selectin-bearing substrates or other cells; Belongs to the LAMP family
*Plekhs1*	Pleckstrin homology domain containing, family S member 1
*Rs1*	Retinoschisis (x-linked, juvenile) 1 (human); Retinoschisin; Binds negatively charged membrane lipids, such as phosphatidylserine and phosphoinositides. May play a role in cell–cell adhesion processes in the retina, via homomeric interaction between octamers present on the surface of two neighboring cells (By similarity). Required for normal structure and function of the retina
*Wap*	Whey acidic protein; Could be a protease inhibitor. May play an important role in mammary gland development and tissue remodeling
*Hiatl1*	Major facilitator superfamily domain containing 14b; Hippocampus abundant transcript-like 1

**Table 4 genes-13-02020-t004:** Functional annotation of Ccl3 protein association network.

Protein	Annotation
Ccl3	C-C motif chemokine 3; Monokine with inflammatory, pyrogenic and chemokinetic properties. Has a potent chemotactic activity for eosinophils. Binding to a high-affinity receptor activates calcium release in neutrophils; Belongs to the intercrine β (chemokine CC) family
Ccr1	Chemokine (c-c motif) receptor 1; C-C chemokine receptor type 1; Receptor for a C-C type chemokine. Binds to MIP-1-α, RANTES, and less efficiently, to MIP-1-β or MCP-1 and subsequently transduces a signal by increasing the intracellular calcium ions level. Responsible for affecting stem cell proliferation
Il6	Interleukin-6; Cytokine with a wide variety of biological functions. It is a potent inducer of the acute phase response. Plays an essential role in the final differentiation of B-cells into Ig- secreting cells Involved in lymphocyte and monocyte differentiation. Acts on B-cells, T-cells, hepatocytes, hematopoietic progenitor cells and cells of the CNS. Required for the generation of T(H)17 cells. Furthermore, acts as a myokine. It is discharged into the bloodstream after muscle contraction and acts to increase the breakdown of fats and to improve insulin resistance.
Ccr5	Chemokine (c-c motif) receptor 5; C-C chemokine receptor type 5; Receptor for a number of inflammatory CC-chemokines including MIP-1-α, MIP-1-β and RANTES and subsequently transduces a signal by increasing the intracellular calcium ion level. May play a role in the control of granulocytic lineage proliferation or differentiation (By similarity)
Csf2	Colony stimulating factor 2 (granulocyte-macrophage); Granulocyte-macrophage colony-stimulating factor; Cytokine that stimulates the growth and differentiation of hematopoietic precursor cells from various lineages, including granulocytes, macrophages, eosinophils and erythrocytes
Cxcl10	C-X-C motif chemokine 10; In addition to its role as a proinflammatory cytokine, may participate in T-cell effector function and perhaps T-cell development; Belongs to the intercrine α (chemokine CxC) family
Tab2	TGF-β-activated kinase 1 and MAP3K7-binding protein 2; Adapter linking MAP3K7/TAK1 and TRAF6. Promotes MAP3K7 activation in the IL1 signaling pathway. The binding of ‘Lys-63’- linked polyubiquitin chains to TAB2 promotes autophosphorylation of MAP3K7 at ‘Thr-187’ (By similarity). Regulates the IL1-mediated translocation of NCOR1 out of the nucleus. Involved in heart development (By similarity)
Tnf	Tumor necrosis factor superfamily, member 2; Tumor necrosis factor; Cytokine that binds to TNFRSF1A/TNFR1 and TNFRSF1B/TNFBR. It is mainly secreted by macrophages and can induce cell death of certain tumor cell lines. It is potent pyrogen causing fever by direct action or by stimulation of interleukin-1 secretion and is implicated in the induction of cachexia, Under certain conditions it can stimulate cell proliferation and induce cell differentiation
Ccl4	Chemokine (c-c motif) ligand 4; C-C motif chemokine 4; Monokine with inflammatory and chemokinetic properties
Il10	Interleukin-10; Inhibits the synthesis of a number of cytokines, including IFN-γ, IL-2, IL-3, TNF and GM-CSF produced by activated macrophages and by helper T-cells
Il6	Interleukin-6; Cytokine with a wide variety of biological functions. It is a potent inducer of the acute phase response. Plays an essential role in the final differentiation of B-cells into Ig- secreting cells Involved in lymphocyte and monocyte differentiation. Acts on B-cells, T-cells, hepatocytes, hematopoietic progenitor cells and cells of the CNS. Required for the generation of T(H)17 cells. Furthermore, acts as a myokine. It is discharged into the bloodstream after muscle contraction and acts to increase the breakdown of fats and to improve insulin resistance.

**Table 5 genes-13-02020-t005:** Functional annotation of Lyz2 protein association network.

Protein	Annotation
Lyz2	Lysozyme C-2; Lysozymes have primarily a bacteriolytic function; those in tissues and body fluids are associated with the monocyte- macrophage system and enhance the activity of immunoagents. Lyz2 is active against a range of Gram-positive and Gram-negative bacteria. More effective than Lyz1 in killing Gram-negative bacteria. Lyz1 and Lyz2 are equally effective in killing Gram- positive bacteria; Belongs to the glycosyl hydrolase 22 family
Lyz1	Lysozyme C-1; Lysozymes have primarily a bacteriolytic function; those in tissues and body fluids are associated with the monocyte- macrophage system and enhance the activity of immunoagents. Lyz1 is active against a range of Gram-positive and Gram-negative bacteria. Less effective than Lyz2 in killing Gram-negative bacteria. Lyz1 and Lyz2 are equally effective in killing Gram- positive bacteria; Belongs to the glycosyl hydrolase 22 family

**Table 6 genes-13-02020-t006:** Functional annotation of Fa2h protein association network.

Protein	Annotation
Fa2h	4-hydroxysphinganine ceramide fatty acyl 2-hydroxylase; Fatty acid 2-hydroxylase; Required for α-hydroxylation of free fatty acids and the formation of α-hydroxylated sphingolipids
Scd1	Stearoyl-coa desaturase (delta-9 desaturase); Acyl-CoA desaturase 1; Stearyl-CoA desaturase that utilizes O(2) and electrons from reduced cytochrome b5 to introduce the first double bond into saturated fatty acyl-CoA substrates. Catalyzes the insertion of a cis double bond at the delta-9 position into fatty acyl-CoA substrates including palmitoyl-CoA and stearoyl-CoA. Gives rise to a mixture of 16:1 and 18:1 unsaturated fatty acids. Plays an important role in lipid biosynthesis. Plays an important role in regulating the expression of genes that are involved in lipogenesis.
Dync1li2	Cytoplasmic dynein 1 light intermediate chain 2; Acts as one of several non-catalytic accessory components of the cytoplasmic dynein 1 complex that are thought to be involved in linking dynein to cargos and to adapter proteins that regulate dynein function. Cytoplasmic dynein 1 acts as a motor for the intracellular retrograde motility of vesicles and organelles along microtubules. May play a role in binding dynein to membranous organelles or chromosomes (By similarity)
Agmo	Alkylglycerol monooxygenase; Glyceryl-ether monooxygenase that cleaves the O-alkyl bond of ether lipids. Ether lipids are essential components of brain membranes (By similarity); Belongs to the sterol desaturase family. TMEM195 subfamily
Aldh3a2	Aldehyde dehydrogenase family 3, subfamily a2; Fatty aldehyde dehydrogenase; Catalyzes the oxidation of long-chain aliphatic aldehydes to fatty acids. Responsible for conversion of the sphingosine 1-phosphate (S1P) degradation product hexadecenal to hexadecenoic acid (By similarity)
Acer3	Alkaline ceramidase 3; Hydrolyzes only phytoceramide into phytosphingosine and free fatty acid. Does not have reverse activity (By similarity)
Asah2	N-acylsphingosine amidohydrolase 2; Neutral ceramidase; Hydrolyzes the sphingolipid ceramide into sphingosine and free fatty acid at an optimal pH of 6.5–8.5. Acts as a key regulator of sphingolipid signaling metabolites by generating sphingosine at the cell surface. Acts as a repressor of apoptosis both by reducing C16-ceramide, thereby preventing ceramide-induced apoptosis, and generating sphingosine, a precursor of the antiapoptotic factor sphingosine 1-phosphate. Probably involved in the digestion of dietary sphingolipids in intestine.
Gba2	Non-lysosomal glucosylceramidase; Non-lysosomal glucosylceramidase that catalyzes the conversion of glucosylceramide (GlcCer) to free glucose and ceramide. Involved in sphingomyelin generation and prevention of glycolipid accumulation. May also catalyze the hydrolysis of bile acid 3-O-glucosides, however, the relevance of such activity is unclear in vivo. Plays a role in central nevous system development (By similarity). Required for proper formation of motor neuron axons (By similarity)
Pla2g6	85/88 kDa calcium-independent phospholipase A2; Catalyzes the release of fatty acids from phospholipids. It has been implicated in normal phospholipid remodeling, nitric oxide-induced or vasopressin-induced arachidonic acid release and in leukotriene and prostaglandin production. May participate in fas mediated apoptosis and in regulating transmembrane ion flux in glucose-stimulated B-cells. Has a role in cardiolipin (CL) deacylation. Required for both speed and directionality of monocyte MCP1/CCL2-induced chemotaxis through regulation of F- actin polymerization at the pseudopods.
1600014C10Rik	Protein C19orf12 homolog; RIKEN cDNA 1600014C10 gene

**Table 7 genes-13-02020-t007:** Functional annotation of Mog protein association network.

Protein	Annotation
Mog	Myelin oligodendrocyte glycoprotein; Minor component of the myelin sheath. May be involved in completion and/or maintenance of the myelin sheath and in cell–cell communication. Mediates homophilic cell–cell adhesion
Zfp57	Krab domain-containing zinc finger protein; Zinc finger protein 57; Transcription regulator required to maintain maternal and paternal gene imprinting, a process by which gene expression is restricted in a parent of origin-specific manner by epigenetic modification of genomic DNA and chromatin, including DNA methylation. Acts by controlling DNA methylation during the earliest multicellular stages of development at multiple imprinting control regions. Required for the establishment of maternal methylation imprints at SNRPN locus. Acts as a transcriptional repressor in Schwann cells.
Aqp4	Aquaporin-4; Forms a water-specific channel. Osmoreceptor which regulates body water balance and mediates water flow within the central nervous system
Plp1	Myelin proteolipid protein; This is the major myelin protein from the central nervous system. It plays an important role in the formation or maintenance of the multilamellar structure of myelin; Belongs to the myelin proteolipid protein family
Mobp	Myelin-associated oligodendrocytic basic protein; Myelin-associated oligodendrocyte basic protein; May play a role in compacting or stabilizing the myelin sheath possibly by binding the negatively charged acidic phospholipids of the cytoplasmic membrane
Mag	Myelin-associated glycoprotein; Adhesion molecule that mediates interactions between myelinating cells and neurons by binding to neuronal sialic acid- containing gangliosides and to the glycoproteins RTN4R and RTN4RL2. Not required for initial myelination, but seems to play a role in the maintenance of normal axon myelination. Protects motoneurons against apoptosis, also after injury; protection against apoptosis is probably mediated via interaction with neuronal RTN4R and RTN4RL2. Required to prevent degeneration of myelinated axons in adults.
Mbp	Myelin basic protein; The classic group of MBP isoforms (isoform 4-isoform 13) are with PLP the most abundant protein components of the myelin membrane in the CNS. They have a role in both its formation and stabilization. The non-classic group of MBP isoforms (isoform 1-isoform 3/Golli-MBPs) may preferentially have a role in the early developing brain long before myelination, maybe as components of transcriptional complexes, and may also be involved in signaling pathways in T-cells and neural cells.
Gfap	Glial fibrillary acidic protein; GFAP, a class-III intermediate filament, is a cell- specific marker that, during the development of the central nervous system, distinguishes astrocytes from other glial cells
Il17a	Interleukin-17A; Ligand for IL17RA. The heterodimer formed by IL17A and IL17F is a ligand for the heterodimeric complex formed by IL17RA and IL17RC (By similarity). Involved in inducing stromal cells to produce proinflammatory and hematopoietic cytokines (By similarity)
Cd4	T-cell surface glycoprotein CD4; Integral membrane glycoprotein that plays an essential role in the immune response and serves multiple functions in responses against both external and internal offenses. In T-cells, functions primarily as a coreceptor for MHC class II molecule:peptide complex. The antigens presented by class II peptides are derived from extracellular proteins while class I peptides are derived from cytosolic proteins. Interacts simultaneously with the T-cell receptor (TCR) and the MHC class II presented by antigen presenting cells (APCs).
Cd209d	C-type lectin domain family 4 member m; CD209 antigen-like protein D; Probable pathogen-recognition receptor. May mediate the endocytosis of pathogens which are subsequently degraded in lysosomal compartments. May recognize in a calcium-dependent manner high mannose N-linked oligosaccharides in a variety of pathogen antigens

**Table 8 genes-13-02020-t008:** Functional annotation of Ugt8a protein association network.

Protein	Annotation
Ugt8a	2-hydroxyacylsphingosine 1-β-galactosyltransferase; Catalyzes the transfer of galactose to ceramide, a key enzymatic step in the biosynthesis of galactocerebrosides, which are abundant sphingolipids of the myelin membrane of the central nervous system and peripheral nervous system; Belongs to the UDP-glycosyltransferase family
Cerk	Ceramide kinase; Catalyzes specifically the phosphorylation of ceramide to form ceramide 1-phosphate. Acts efficiently on natural and analog ceramides (C6, C8, C16 ceramides, and C8-dihydroceramide), to a lesser extent on C2-ceramide and C6-dihydroceramide, but not on other lipids, such as various sphingosines (By similarity)
Degs2	Sphingolipid delta(4)-desaturase/C4-monooxygenase DES2; Bifunctional enzyme which acts as both a sphingolipid delta(4)-desaturase and a sphingolipid C4-monooxygenase; Belongs to the fatty acid desaturase type 1 family. DEGS subfamily
Cers3	Ceramide synthetase; Ceramide synthase 3; Has (dihydro)ceramide synthesis activity with relatively broad substrate specificity, but a preference for C18:0 and other middle- to long-chain fatty acyl-CoAs. It is crucial for the synthesis of very long-chain ceramides in the epidermis, to maintain epidermal lipid homeostasis and terminal differentiation (By similarity)
Smpd4	Sphingomyelin phosphodiesterase 4; Catalyzes the hydrolysis of membrane sphingomyelin to form phosphorylcholine and ceramide
Sgpp2	Sphingosine-1-phosphate phosphotase 2; Sphingosine-1-phosphate phosphatase 2; Has specific phosphohydrolase activity towards sphingoid base 1-phosphates. Has high phosphohydrolase activity against dihydrosphingosine-1-phosphate and sphingosine-1-phosphate (S1P) in vitro. May play a role in attenuating intracellular sphingosine 1-phosphate (S1P) signaling. May play a role in pro-inflammatory signaling (By similarity)
Ugcg	Udp-glucose ceramide glucosyltransferase; Ceramide glucosyltransferase; Catalyzes the first glycosylation step in glycosphingolipid biosynthesis, the transfer of glucose to ceramide. May also serve as a “flippase” (By similarity)
Cers2	Ceramide synthetase; Ceramide synthase 2; Suppresses the growth of cancer cells. May be involved in sphingolipid synthesis (By similarity)
Sgms1	Phosphatidylcholine:ceramide cholinephosphotransferase 1; Sphingomyelin synthases synthesize the sphingolipid, sphingomyelin, through transfer of the phosphatidyl head group, phosphatidylcholine, on to the primary hydroxyl of ceramide. The reaction is bidirectional depending on the respective levels of the sphingolipid and ceramide. Golgi apparatus SMS1 directly and specifically recognizes the choline head group on the substrate, requiring two fatty chains on the choline-P donor molecule in order to be recognized efficiently as a substrate. Major form in macrophages.

## Data Availability

The data presented in this study are available on request from the corresponding author. The information regarding the context sequence of probes used in this study is available in Appendix A section of this manuscript.

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
