# Peer review of "Gene Expression Profile in the Sandhoff Mouse Brain with Progression of Age"

_genes, 2022, doi:10.3390/genes13112020_

Round 1
Reviewer 1 Report
The manuscript titled ""Gene expression profile in the Sandhoff mouse brain with pro-2 gression of age" by Singh et al. is an important piece of work exploring the expression profile of selected immune and myelination associated transcripts (Wfdc17, Ccl3, Lyz2, Fa2h, Mog and Ugt8a) at 5-, 10- and 16-weeks, 13 representing young, pre-symptomatic and late stages of the Sandhoff Disease mice. While the findings of this work are important, it is pertinent to note that the results should be interpreted with cautions owing to the exploration of a limited number of genes in the disease.
The authors are requested to address the following issues:
1. The authors should clearly highlight the need for exploring only 6 genes. Are there other genes related to immune and myelination that could have been explored? If not, can the authors explain why the traditionally explored genes of immunological pathways are not relevant in this context. If yes, the authors should clarify why that they not been included for the study.
2. Functional association network analysis identifies several relevant proteins. Exploration of the protein expression (or the mRNA expression) some of the relevant genes may be helpful for the readers to understand how these networks are impacted with the changes in the expression levels of the genes studied here.
3. Mention the PCR cycling conditions in the methodology.
4. The quality of the RNA should be mentioned in the methodology
Author Response
Thank you for reviewing the manuscript. Please see the attached file for our responses.

Reviewer 2 Report
This is interesting manuscript with the purpose to explore the expression profile of selected immune and myelination associated transcripts (Wfdc17, Ccl3, Lyz2, Fa2h, Mog and Ugt8a) at 5-, 10- and 16-weeks. All sections of the manuscript are clear and concise; however, I suggest to consider next few minor comments to improve the manuscript:
- Line 76: In qPCR section, please explain under what laboratory conditions was performed the RT-PCR.
- Line 84: Please explain, what does Heterozygote means?
- Line 87: Please explain in this section, which were the genes used to construct each individual network?, or, where these genes come from?
- Line 106: In Results section, I suggest to use verbs in past tense instead of present tense. Also, please replace “(p-value < 0.05)” by “(P < 0.05)”.
- Lines 216-217: This sentence appears to belong to Results section.
- Line 312: In References section, usually only the first letter at the beginning of the article title should be capitalized.
Author Response

(The authors gave the same response as above.)
